# Position: Multiplicity is an Inevitable and Inherent Challenge in Multimodal Learning

**Sanghyuk Chun** [1]  **Olga Russakovsky** [1]

## Abstract

Multimodal learning has seen remarkable progress, particularly with large-scale pre-training across various modalities. Most current approaches are built on the assumption of a deterministic one-to-one alignment between modalities. However, this oversimplifies real-world multimodal relationships, where their nature is inherently many-to-many. The many-to-many property, or *multiplicity*, is not a side-effect of noise or annotation error, but an inevitable outcome of intra-modal variability, representational asymmetry, and task-dependent ambiguity in multimodal tasks. We argue that multiplicity is a fundamental bottleneck that affects all stages of the multimodal learning pipeline: from data construction to model training and evaluation benchmarks. By formalizing its causes and consequences, we demonstrate how ignoring multiplicity leads to training uncertainty, unreliable evaluation, and degraded dataset quality. This position paper calls for new research directions on multimodal learning, including multiplicity-aware learning frameworks and dataset construction and evaluation protocols.

## 1. Introduction

Multimodal learning has emerged as a foundation in modern machine learning, showing recent breakthroughs in tasks involving vision, language, audio, action, and beyond (Radford et al., 2021; Jia et al., 2021; Li et al., 2022a; Liu et al., 2023; Elizalde et al., 2023; Ahn et al., 2023; Driess et al., 2023; Kim et al., 2024a). The rise of large-scale pre-training has significantly expanded what these systems can achieve. However, this success relies on a fragile, simplifying assumption: that mappings across modalities are *one-to-one*. Whether for contrastive pre-training or retrieval-based evaluation, each instance in one modality is assumed to correspond to exactly one correct counterpart in another, *e.g.*, one image to one caption. This one-to-one alignment assumption is fundamentally misaligned with the nature of real-world multimodal data. In practice, the relationship between modalities is inherently many-to-many, *e.g.*, an image can be described by multiple captions and vice versa, a property we define as **"multiplicity"**, the existence of multiple plausible correspondences between modalities.

This position paper argues that **multiplicity is an inevitable and inherent challenge in multimodal learning, and multimodal learning should be reframed around multiplicity.** Throughout the paper, we will show how multiplicity affects the entire multimodal learning pipeline, from data construction, training (*e.g.*, contrastive pre-training), to retrieval-based evaluation. Multiplicity is not a simple noise or side-effect, but a fundamental characteristic.

The roots of multiplicity are manifold and diverse. First, current multimodal dataset construction pipelines capture only a **sparse sampling of the potential correspondence space**, which grows quadratically with dataset scale. Second, there exists **intra-modal variability**: multiple instances in one modality correspond to the same semantic concept. For example, as shown in Figure 1 (a), a single concept (*e.g.*, cat) can be instantiated in diverse ways within an image modality. Third, there are **asymmetries in information density and representation mechanisms** (*e.g.*, dense image exhaustively captured by photographic sensors versus sparse linguistic descriptions with selectively chosen concepts by humans). The same modality item can be interpreted in multiple valid ways when expressed in the other modality, and it makes complete and symmetric alignment infeasible. Ambiguity in what "counts" as a corresponding item leads to multiple valid alignments (See Figure 1 (b)). Finally, **the definition of correspondence depends on task objectives or context**. Different tasks demand different alignment notions, *e.g.*, for vision-language tasks, should an image be aligned to a caption describing its category, its background, its future implication, or its narrative framing? For audio-visual tasks, should a sound be aligned to on-screen actions, ambient context, or narrative tone? There is no single "true" counterpart. The set of valid correspondences varies by

[1]Princeton University. Correspondence to: Sanghyuk Chun <sanghyukc@princeton.edu>.

*Proceedings of the $43^{rd}$ International Conference on Machine Learning*, Seoul, South Korea. PMLR 306, 2026. Copyright 2026 by the author(s).

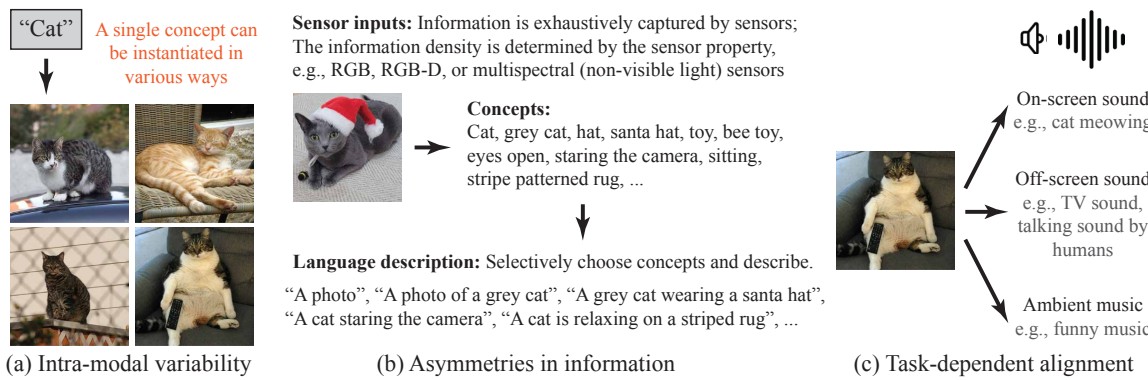

(a) Intra-modal variability      (b) Asymmetries in information      (c) Task-dependent alignment

*Figure 1.* **How does multiplicity occur?** The source of multiplicity in multimodal datasets is diverse.

purpose, introducing conditional multiplicity as shown in Figure 1 (c). Namely, there is no single "truly corresponding pair" for a given instance; it depends on how we define the task. Section 2 will discuss more details of the sources of multiplicity.

This multiplicity is unavoidable in practice for multimodal tasks. Unfortunately, the space of potentially valid cross-modal correspondences expands rapidly with scale, making it infeasible in practice to enumerate or verify all plausible matches. Therefore, cross-modal supervision is necessarily sparse, and multiplicity can induce false negatives (FNs) that affect both training and evaluation. We formalize this notion in Section 2 and discuss its implications throughout the pipeline. Considering these problems, multiplicity should be carefully considered during dataset construction, as design choices at this stage can either preserve or suppress the many-to-many nature of modality relationships.

## 2. Multiplicity: An inherent challenge

**Definition.** Let $\mathcal{R} \subseteq \mathcal{X} \times \mathcal{Y}$ denote valid cross-modal relations between two modalities $\mathcal{X}$ and $\mathcal{Y}$ (*e.g.*, vision-language (Radford et al., 2021), audio-visual (Elizalde et al., 2023)). Note that we assume two modalities for simplicity, but this definition can be easily extended to $n$ modalities, such as vision-language-action (Ahn et al., 2023; Driess et al., 2023) and video-language-audio (Jeong et al., 2025), $\mathcal{R} \subseteq \mathcal{X}_1 \times \mathcal{X}_2 \times \ldots \times \mathcal{X}_n$. Standard practice presumes one-to-one correspondence, *i.e.*, $|\{y \in \mathcal{Y} : (x, y) \in \mathcal{R}\}| = 1$ for all $x \in \mathcal{X}$ (and symmetrically, $|\{x \in \mathcal{X} : (x, y) \in \mathcal{R}\}| = 1$ for all $y \in \mathcal{Y}$). **Multiplicity** (or many-to-many correspondence) occurs when there exists some $x \in \mathcal{X}$ such that $|\{y \in \mathcal{Y} : (x, y) \in \mathcal{R}\}| > 1$ (or symmetrically, some $y \in \mathcal{Y}$ such that $|\{x \in \mathcal{X} : (x, y) \in \mathcal{R}\}| > 1$).

In the worst case, $|\mathcal{R}|$ can be as large as $|\mathcal{X}| \cdot |\mathcal{Y}|$, *i.e.*, the space of valid relations can grow rapidly when both modalities scale. Furthermore, cross-modal supervision is neces-

sarily sparse: even when some correspondences are labeled as positives, additional plausible positives typically remain unobserved among pairs treated as "negatives." In practice, multimodal datasets typically record only a sparse set of positive pairs and treat unobserved pairs as negatives, so valid but unannotated correspondences become false negatives (FNs). This property introduces challenges throughout the multimodal learning pipeline (we will discuss more details in later sections). This makes multiplicity a first-order concern when scaling multimodal datasets. On the other hand, unimodal tasks (*e.g.*, single or multi-labeled classification with fixed label sets) do not directly suffer from the same issue. In Appendix A, we compare unimodal tasks with multimodal tasks for additional intuition.

We characterize origins of multiplicity in real-world data with three primary sources: intra-modal variability, asymmetry between modalities, and task-dependent alignment.

**Property 1. Intra-modal variability.** Assume a data generation process (*e.g.*, structural causal models (Pearl et al., 2000)) from the underlying "concepts" to the actual data. For example, consider visual and textual instances generated from concepts "grey cat", "santa hat", and "striped rug" (*e.g.*, Figure 1 (b)). This generation process is inherently stochastic, with no uniquely determined instance. As a result, each modality realizes the concepts in various shapes, *e.g.*, images with slightly different views or backgrounds, and diverse captions describing the same situation (See Figure 1 (a)). Namely, if there exist two semantically similar multimodal pairs with overlapping concepts $(x_1, y_1)$ and $(x_2, y_2)$, their cross-relationships $(x_1, y_2)$ and $(x_2, y_1)$ should also be treated as valid positives even though they are treated as negative in the dataset. This problem becomes significant when we restrict the possible objects in the datasets and the data format (*e.g.*, COCO Caption (Chen et al., 2015) is built upon COCO (Lin et al., 2014) images of 80 common objects).

**Property 2. Asymmetry between modalities.** Modalities differ in how they encode and express information. For example, a photograph exhaustively records visual details, while a human-written caption selectively reflects only a few salient concepts. Although the same concept may appear in both modalities in varied forms, their information density differs significantly, especially in text, which is based on human cognition rather than sensor-based input. Cognition theories, such as dual-coding theory (Paivio, 1990), suggest that the mind processes information along verbal and non-verbal systems. When a person writes "a grey cat wearing a Santa hat" the verbal code is followed by a private visual image that may include additional details (background, action) never lexicalized. Different annotators, therefore, generate distinct but equally valid sentences for the same scene, and a single sentence can evoke multiple mental images, immediately yielding many-to-many alignments. Even sensor inputs have different information density by the choice of the sensor. For example, visual inputs captured by RGB, RGB-D, non-visible light, video camera, and motion sensors have different information from each other; the same scene will be expressed differently by the sensors.

**Property 3. Task-dependent alignment.** What counts as a correct alignment often depends on the task (Chun et al., 2026). For example, in vision-language tasks, should a caption describe only the main object in the image (Chen et al., 2015)? Should it exhaustively describe all the local visual information (Pont-Tuset et al., 2020)? Infer what happened before and what happens next (Park et al., 2020)? In audio-visual settings in Figure 1 (c), the notion of alignment could range from on-screen sounds (*e.g.*, cat meowing sound) (Chen et al., 2020), off-screen sounds (*e.g.*, TV sound), talking speech following lip movement (Nagrani et al., 2017), or ambient sounds (*e.g.*, background music or foley effects) (Owens et al., 2016). Namely, the definition of a "positive" pair is ambiguous, context-sensitive, and task-dependent; a pair that is positive under one task definition may be irrelevant or even negative under another (*e.g.*, ambient sounds could be negative if we only focus on on-screen sounds).

The above sources are conceptually distinct, but they induce the same practical consequence: sparse one-to-one annotations fail to capture the full set of valid cross-modal correspondences (See Appendix C.3 for related discussions). We next summarize empirical evidence from prior work showing that this mismatch appears in multimodal tasks.

**Empirical evidence.** Recent studies implicitly support the existence of multiplicity and its potential effects in multimodal tasks, when we over-rely on the one-to-one assumption. For example, Chun et al. (2022) empirically showed that a popular image-caption dataset misses many valid matches. This study showed that COCO Caption (Chen et al., 2015) contains many redundant captions, which results in FNs in the dataset; the average number of positive images (captions) for each caption (image) is 8.5 (17.9) rather than the original 1 (5) in the dataset. Chun et al. (2022) also showed that re-evaluating VLMs with corrected associations can noticeably change model rankings (*e.g.*, showing 0.47 Kendall's ranking coefficient). In summary, one-to-one benchmarks can become unreliable under multiplicity.

On the training side, Chun (2024) showed that the existing training methods assuming a strong one-to-one correspondence can break down as model scale increases (*e.g.*, achieving 40.0 mAP@R on ViT-B/32 but 20.2 mAP@R on ViT-L/14), while approaches that account for multiplicity remain much more stable (40.1 mAP@R on ViT-B/32, 42.1 mAP@R on ViT-L/14). In other words, if the effect of FNs (due to multiplicity) becomes more significant, conventional training strategies can fail.

Kim et al. (2024b) provided a relevant example on the dataset side. They improved training dataset quality by filtering out underspecified image-text pairs (*i.e.*, pairs with a higher degree of multiplicity). This supports that multimodal dataset quality is highly affected by its multiplicity.

Overall, the nature of multimodal correspondences is many-to-many, and the existing works support that multiplicity is an important structural challenge in multimodal learning. In the next sections, we examine how this multiplicity impacts data collection, training, and evaluation in more detail.

## 3. Multiplicity in training

### 3.1. How does multiplicity induce ambiguity in training?

Mainstream multimodal architectures (Lu et al., 2019; Radford et al., 2021; Kim et al., 2021; Zhai et al., 2023) assume a one-to-one mapping, *i.e.*, each instance is encoded into a unique representation vector. However, multimodal inputs are inherently polysemous: a single instance can correspond to multiple valid interpretations or alignments, each deserving a distinct representation. If we assume an ideal dataset that annotates all plausible matches as positives, this multiplicity cannot be faithfully captured by one-to-one encodings. For example, as shown in Figure 2 (a), a cat image should simultaneously match multiple captions with different meanings, which is fundamentally impossible by a one-to-one mapping. This introduces **input ambiguity**, or aleatoric uncertainty; an input can be represented variously.

In practice, most multimodal datasets (Pont-Tuset et al., 2020; Changpinyo et al., 2021; Desai et al., 2021; Schuhmann et al., 2021; 2022) provide sparse one-to-one annotations because exhaustively annotating all plausible correspondences is infeasible. Importantly, this is not inherently

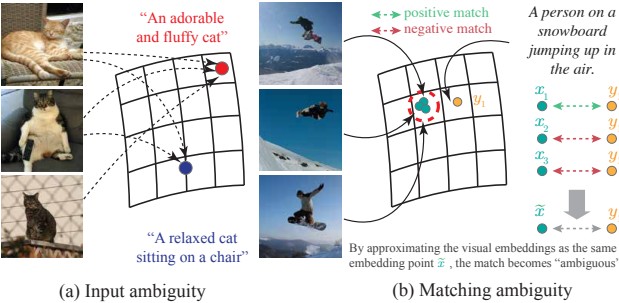

(a) Input ambiguity       (b) Matching ambiguity

*Figure 2.* **Multiplicity induces ambiguity.** (a) If we have an ideal dataset consists of the full pairwise annotations, an input should correspond to multiple instances from the other modality. The current one-to-one paradigm cannot handle this. (b) In practice, we have sparsely annotated pairwise annotations: each input only corresponds to one instance. In this case, multiplicity introduces a new uncertainty, named matching ambiguity.

flawed: a single caption or counterpart can be a valid sample from the set of acceptable correspondences. The problem arises when contrastive training or retrieval-style evaluation treats this sparse annotation as exhaustive, so that all unannotated but plausible pairs are used as negatives. Under this use, **false negatives (FNs)** naturally emerge.

While each input has only one "ground truth" (hence, the input-level ambiguity is collapsed in supervision), the ambiguity still exists at the level of pairwise relationships. In this case, models suffer from **matching ambiguity**: a given multimodal correspondence can be either positive or negative. This is another form of aleatoric uncertainty, not over the inputs themselves but over their cross-modal alignments. We examine how matching ambiguity arises.

As discussed in Section 2 *intra-modal variability*, multiple semantically similar items often exist within each modality. When we approximate such items (*e.g.*, images of the same object in different views, or captions describing the same scene with varying detail) into a single representation (by assuming that the encoder maps similar inputs into a very close and almost the same space), the resulting cross-modal matching becomes intrinsically ambiguous. Figure 2 (b) illustrates the overview.

Formally, suppose $\{x_1, x_2, \ldots, x_K\} \subset \mathcal{X}$ are semantically equivalent inputs, approximated as a single representative $\tilde{x}$. Let $\{y_1, \ldots, y_K\} \subset \mathcal{Y}$ be their corresponding instances from another modality and the annotated positive relations are $\{(x_i, y_i) \in \mathcal{R} \mid i = 1, \ldots, K\}$. Assume we randomly sample $(x_i, y_j)$ from the mini-batch $\{(x_i, y_i) \mid i = 1 \ldots K\}$. Then, the matching label $m$ between $\tilde{x}$ and $y_j$ becomes a stochastic variable: $m(\tilde{x}, y_j) = m(x_i, y_j) = 1$ if $i = j$ and $0$ otherwise. If we assume that $y$ is approximated as $\tilde{y}$, the probability of positive matching between $\tilde{x}$ and $\tilde{y}$ is $1/K$.

**Recap of current multimodal learning training algorithms.** Modern multimodal learning heavily relies on training objectives that assume well-defined, one-to-one multimodal correspondences. Approaches such as triplet loss with hard negative mining (Faghri et al., 2018; Chen et al., 2021), contrastive learning (Radford et al., 2021; Zhai et al., 2023), pairwise matching (Lu et al., 2019; Kim et al., 2021), and instruction tuning (Liu et al., 2023) all follow a similar principle: bring positive pairs closer while pushing negatives apart. They work under the assumption that each input has a single, correct counterpart in the other modality. When this assumption fails due to the input ambiguity or matching ambiguity, the model is penalized for preserving the correct semantic structure. This misalignment leads to undesirable outcomes: (1) distances between semantically compatible items become exaggerated, and (2) models may overfit to arbitrary choices among positive matches by disrupting the stability of gradient signals, especially when only one ground-truth is used in training.

**Settings.** Let $x \in \mathcal{X}$ and $y \in \mathcal{Y}$ be items from two modalities. Each mini-batch contains $N$ instances, with $N$ annotated positive pair $\{(x_i, y_i) \mid i = 1 \ldots N\}$. We suppose that the first $K$ pairs form a semantically equivalent cluster such that each $x_i$ ($i = 1, \ldots, K$) has $K$ equally valid matches $\mathbf{y}_+ = \{y_1, \ldots, y_K\}$ in the mini-batch. In this case, the total number of true positive relations is $N - K + K^2$, whereas the dataset may only annotate $N$ of them as positive, leaving $K^2 - K$ relations unobserved and thus treated as negatives, *i.e.*, FNs. Let $f(x)$ and $g(y)$ denote the normalized embedding by encoders $f$ and $g$.

**Contrastive loss.** Let $p_j := \frac{\exp(f(x_1)^\top g(y_j))}{\sum_{k=1}^{N} \exp(f(x_1)^\top g(y_k))}$, the softmax probability that $x_1$ and $y_j$ are matched. For $x_1$, the original contrastive loss (*i.e.*, only considering $N$ positives) is defined by $\mathcal{L}_{\text{sparse}} = -\log p_1$ and its gradient w.r.t. $f(x_1)$ is $\nabla_{f(x_1)} \mathcal{L}_{\text{sparse}} = \sum_{j=1}^{N} p_j g(y_j) - g(y_1)$; this gradient becomes $0$ when $p_1 = 1$, pushing $f(x_1)$ and $g(y_1)$ closer while pulling $f(x_1)$ away from all other $g(y_j)$. However, if we suppose that $x_1$ has $K > 1$ actually valid positives but unannotated $\mathbf{y}_+$. Then, the gradient pulls $f(x_1)$ away from $g(y_+)$ despite their semantic similarity.

In contrast, an ideal loss considering all positives uniformly account for all $K$ positives: $\mathcal{L}_{\text{ideal}} = \sum_{j=1}^{K} (-\frac{1}{K} \log p_j)$. Let $p_j^* = \frac{1}{K}$ for $j = 1 \ldots K$ and $0$ otherwise. Then, the gradient of $\mathcal{L}_{\text{ideal}}$ w.r.t. $x_1$ becomes: $\nabla_{f(x_1)} \mathcal{L}_{\text{ideal}} = \sum_{j=1}^{N} p_j g(y_j) - \sum_{j=1}^{N} p_j^* g(y_j)$; this gradient becomes $0$ when $p_j = \frac{1}{K}$ for all $j = 1 \ldots K$. More specifically, the discrepancy between the actual and ideal gradients becomes $\sum_{j=1}^{N} (p_j - p_j^*) g(y_j)$. This mismatch makes the distance between $x_1$ and its plausible matching $y_+$ larger; despite $x_1$ and $y_+$ being actually positive, there exists a gap between

the two modalities, which can lead to the modality gap (Liang et al., 2022). As $K$ increases, this mismatch amplifies, leading to slower convergence (Huynh et al., 2022) and greater semantic fragmentation in the learned embeddings.

**Hard negative mining (HNM).** HNM is a widely used technique in multimodal metric learning that focuses on the most challenging negatives (Faghri et al., 2018; Chen et al., 2021). However, it is particularly vulnerable to FNs when their similarity $f(x_i)^\top g(y_+)$ is comparable to that of the true positive $f(x_i)^\top g(y_1)$. In this case, HNM aggressively pushes $f(x_i)$ away from $g(y_+)$, often more strongly than contrastive learning, resulting in a distorted embedding space that violates semantic consistency.

**Extension to generative and instruction-tuned VLMs.** The same issue arises in generative multimodal training, such as instruction tuning of VLMs (Liu et al., 2023), where the model learns a conditional distribution $p_\theta(y \mid x, t)$ over textual outputs $y$ given multimodal context $(x, t)$. In practice, datasets provide only a single reference output $y^\star$ per context, even though there may exist a set of valid responses $\mathcal{Y}_+(x, t)$. Maximizing $\log p_\theta(y^\star \mid x, t)$ therefore implicitly suppresses probability mass on other valid but unobserved responses, yielding an under-dispersed distribution and reduced output diversity. In this sense, one-reference supervision in generation plays an analogous role to FNs in contrastive learning: it penalizes plausible alternatives that are not annotated. This matters beyond retrieval. As an example, Sterz et al. (2025) suggested that strong MLLMs (*e.g.*, GPT-4 (Achiam et al., 2023), Gemini (Team et al., 2023)) still struggle once evaluation explicitly allows multiple correct answers, *i.e.*, when multiplicity is introduced in the benchmark.

### 3.2. Current attempts and future directions

Despite its significance, the impact of multiplicity during training remains underexplored, particularly in large-scale settings such as vision-language embeddings (Radford et al., 2021; Zhai et al., 2023) or multimodal LLMs (Liu et al., 2023). Several attempts have been made using a smooth loss (Byun et al., 2024), pseudo-label (Li et al., 2022b; Chun, 2024), or mixed label (Chun, 2024) using mixing augmentations (Zhang et al., 2018a; Yun et al., 2019), but their impacts are yet limited. While smaller-scale datasets (Chen et al., 2015) have been used to study the issue, existing approaches show limited scalability and generalizability.

One line of work treats multimodal alignments as noisy correspondence (NC) (Huang et al., 2021) (*i.e.*, considering that a specific portion of annotations are noisy), leveraging techniques from learning with noisy labels (Song et al., 2022). However, this approach has shown limited success in large-scale settings; for example, Chun (2024) reported

that this direction shows negligible benefits over standard contrastive learning. Moreover, architectures and training objectives for NC still assume the one-to-one mapping, limiting in representing inherent input ambiguity. Nonetheless, rethinking a multimodal task with sparsely annotated many-to-many pairwise datasets as learning with noisy labels or positive-unlabeled learning (Bekker & Davis, 2020) will be an interesting future research direction.

Another direction focuses on producing multiple embeddings, rather than a single embedding for each instance (Song & Soleymani, 2019; Kim et al., 2023), where an instance is mapped to a set of representations to capture polysemous context, and similarity is defined via set-to-set relationships. This method assumes a fixed number of latent components per input (*e.g.*, two embeddings for each instance), each intended to capture a distinct concept. While this direction conceptually fits with both input uncertainty and matching uncertainty, it lacks flexibility when there exists more concepts than the pre-defined components and remains unproven at scale. Conceptually, mixture-of-experts (MoE) (Shazeer et al., 2017; Dai et al., 2024) can be an alternative of this direction, but the link between MoE and multiplicity is still underexplored.

Probabilistic embeddings (Chun et al., 2021; Upadhyay et al., 2023; Li et al., 2023a; Chun, 2024; Baumann et al., 2026; Chun et al., 2025; Chun & Yun, 2025) offer a more scalable alternative by modeling each instance as a probabilistic distribution, thereby naturally capturing uncertainty in both representation and alignment. This family of methods has been extended to large-scale VL models (Chun et al., 2025; Chun & Yun, 2025), achieving performance competitive with CLIP. However, the empirical gains from probabilistic modeling remain modest in real-world applications, and their practical utility is still subject to debate.

Despite these directions, the field lacks a unified framework that systematically addresses multiplicity in multimodal training. We encourage rethinking multimodal training, including architecture, representation space, and training objectives, with the inherent input and matching uncertainties.

## 4. Multiplicity in evaluation

### 4.1. Multiplicity makes benchmarks unreliable

Most multimodal benchmarks rely on sparse one-to-one annotations as a practical simplifying assumption, given the datasets and annotation tools available. However, evaluation metrics often treat this simplification as exhaustive ground truth, making them vulnerable to multiplicity. More specifically, multimodal models are often evaluated by one of the following approaches: (1) zero-shot evaluation by defining tasks via modality-specific information; (2) cross-modal retrieval, where the goal is to retrieve corresponding

items across modalities (*e.g.*, image-to-text, text-to-audio); and (3) evaluation of generated outputs, such as captioning, audio synthesis, or robotic action plans. Multiplicity undermines benchmark reliability in two ways: it transforms valid, unannotated correspondences into false negatives (FNs) and creates a disconnect between evaluation metrics and human relevance. Cross-modal retrieval and generation evaluation are particularly vulnerable to multiplicity because they rely on sparse pairwise annotations or limited references. Zero-shot evaluation can be relatively more robust to this problem, but we still need a careful task definition.

Zero-shot evaluation defines tasks using modality-specific information (mostly based on textual description). For example, language-driven models perform zero-shot classification tasks by treating class labels as textual descriptions and performing classification via cross-modal similarity (Radford et al., 2021). As another example, vision-language-action (VLA) models perform tasks based on text instruction sets, and evaluate the plan success rate (Ahn et al., 2023). This paradigm relaxes the pre-defined and fixed task condition by modality-specific information (mostly based on text descriptions, but not mandatory to be language, *e.g.*, task can be defined by audio, such as speech (Lee et al., 2025)). While zero-shot classification can sometimes avoid the pitfalls of multiplicity, this is largely contingent on how the label space is constructed. If class labels are distinct and mutually exclusive, the evaluation remains stable. However, in the case of taxonomic hierarchies (*e.g.*, "Cat" vs. "Russian Blue") or lexical ambiguity (*e.g.*, "laptop computer" vs. "notebook computer" in ImageNet classes (Kisel et al., 2025)), the presence of multiple valid labels per instance challenges the assumption of single-label correctness (Beyer et al., 2020; Shankar et al., 2020; Yun et al., 2021). To make zero-shot evaluation more reliable, the task should be carefully designed considering multiplicity.

In contrast, cross-modal retrieval is directly and severely impacted by multiplicity. Multiplicity inherently leads to false negatives, while most datasets assume a single correct target for each query. However, as the space of plausible correspondences grows rapidly with scale, it is infeasible to densely annotate all the possible matches between two modalities. Specifically, when a dataset is built upon limited objects (*e.g.*, 80 common objects) and a fixed format (*e.g.*, describing the main object), cross-modal retrieval results are often unreliable. For instance, the ECCV Caption benchmark (Chun et al., 2022) demonstrates that a significant portion of COCO Caption (Chen et al., 2015) treated as negatives are in fact semantically correct for human annotators ($\approx \times 4.4$ positive matches than the original dataset). Furthermore, if we consider multiple positives for each query, the evaluation metric also matters in cross-modal retrieval benchmarks; the convention is Recall@K (R@K), but it is often misaligned with human judgments when multiple

relevant matches exist.

Most cross-modal retrieval benchmarks assume that each query corresponds to exactly one positive target. This leads to the widespread use of R@K, which simply check whether the positive appears within the top-K retrieved items. Prior work shows that single-positive R@K can be misleading under multiplicity, while ranking-sensitive metrics (*e.g.*, mAP@R) better reflect overall ranking quality and correlate more strongly with human preference (Musgrave et al., 2020; Chun et al., 2022). We provide a detailed discussion and examples in Appendix B.

Finally, evaluating generated outputs under multiplicity introduces a different set of challenges. Generative tasks are inherently open-ended, and the space of plausible outputs is vast and diverse (Lee et al., 2023). Traditional automatic metrics evaluate generated outputs by comparing them to a limited set of reference outputs (Heusel et al., 2017), typically using surface-level measures like n-gram overlap (Papineni et al., 2002; Lin, 2004; Banerjee & Lavie, 2005) or latent-level comparison (Zhang et al., 2018b). Image captioning is a canonical example of this issue: many descriptions can be valid for the same image, and recent surveys discuss how evaluation metrics have attempted to handle this diversity across lexical, semantic, and MLLM-based criteria (Sarto et al., 2025). However, limited references still fail to cover many semantically appropriate generations. For example, "a grey cat in the house" and "a Russian Blue playing inside" are different phrasing but equally valid; automatic metrics cannot distinguish them. In this setting, multiplicity leads to systematic underestimation of model quality, as diverse but valid outputs are treated as incorrect. As a result, evaluation can systematically underestimate both correctness and diversity. This highlights a fundamental limitation of current generation-based evaluation protocols in the presence of multimodal ambiguity.

### 4.2. Current attempts and future directions

The most direct way to address multiplicity in evaluation is to exhaustively annotate all plausible cross-modal pairs. However, this is infeasible in practice due to the quadratic growth in the number of possible correspondences. Instead, existing work has explored two main directions.

The first is to automatically identify additional positives using side information such as attributes or semantic similarity. For instance, Chun et al. (2021) introduced densely annotated retrieval benchmarks on CUB (Wah et al., 2011) and COCO (Lin et al., 2014) datasets with fine-grained attributes and object labels. This approach helps mitigate FNs and enables the use of precision metrics, thanks to multiple positives per query. However, it may suffer from false positives, especially when captions refer to scene elements not captured by the predefined object labels. As another example,

Wray et al. (2021) considered semantic similarity proxies computed on captions (*e.g.*, bag-of-words or part-of-speech overlap) for a more reliable video retrieval evaluation. This highly relies on the quality of the similarity proxies.

The second direction is to manually annotate a reduced set of candidate pairs, selected via automatic methods (Parekh et al., 2021; Chun et al., 2022). For example, Chun et al. (2022) used five different retrieval models to select up to 25 candidate matches per query. Human annotators then verified whether each candidate was a true match. This is significantly cheaper than full annotation, but still has a risk of FNs if valid matches are omitted during candidate selection. Also, the scalability of this approach is not promising.

While multiplicity has been relatively actively discussed in retrieval evaluation, its implications are even less explored in other settings. In generation-based evaluation, human judgment remains the de facto standard to handle semantic diversity, as automatic metrics are often unreliable under open-ended outputs. Although human evaluation better reflects real-world diversity, the lack of scalable and reliable automatic metrics continues to slow progress. More broadly, this suggests that evaluation should move beyond single-reference correctness and explicitly assess set- or distribution-level fidelity, *i.e.*, whether a model can capture a range of valid outputs while avoiding invalid ones.

In zero-shot tasks, multiplicity can be partially addressed with ideas from classification. Previous works (Beyer et al., 2020; Shankar et al., 2020; Yun et al., 2021) have proposed rethinking single-label benchmarks, such as ImageNet (Russakovsky et al., 2015), as multi-label tasks or refining label sets to reduce ambiguity. Similar strategies could be applied to zero-shot multimodal evaluation, such as revisiting prompts or category definitions in benchmarks. More generally, because relevance is task-dependent, benchmark design should explicitly specify what notion of correspondence is being evaluated (*e.g.*, object identity vs. attributes vs. narrative context), rather than leaving it implicit.

Ultimately, a faithful evaluation framework must explicitly account for the many-to-many nature of multimodal relationships, both in how relevance is defined (task-dependent correspondence) and how performance is measured (crediting multiple valid outputs rather than a single target).

## 5. Multiplicity in dataset construction

### 5.1. Multiplicity and multimodal dataset quality

Recent studies have shown that multimodal model performance is closely tied to both model and dataset scale (Cherti et al., 2023). As traditional dataset construction is labor-intensive (*e.g.*, manual captions written by human annotators (Chen et al., 2015)), recent approaches focus on collecting large-scale but noisy multimodal pairs (typically crawled from the web) and filtering them to remove low-quality examples (Changpinyo et al., 2021; Schuhmann et al., 2022). Specifically, the existing dataset construction process concentrates on "alignment", measured by a large-scale pre-trained model (Gadre et al., 2024; Maini et al., 2024; Fang et al., 2024). For example, large-scale image-text datasets, such as LAION-5B (Schuhmann et al., 2022), discard image-text pairs whose CLIP similarity is smaller than a pre-defined threshold. This heuristic has become a rule-of-thumb for scalable multimodal dataset construction.

However, as dataset size increases, the strategy that discards or keeps pairs with CLIP similarity may not be enough. Adding a new multimodal pair can introduce many additional plausible correspondences with existing instances and can influence the multiplicity structure of the entire dataset. For example, underspecified instances (*e.g.*, "photo" or "a person is standing") tend to align with a large number of items (*e.g.*, all general photos or human figures), amplifying multiplicity (*i.e.*, increasing the number of plausible matches), leading to input- and matching-ambiguity as discussed in Section 3. Several studies attempted to avoid this challenge by training multimodal models solely with unimodal datasets (*e.g.*, text-only training) (Nukrai et al., 2022; Gu et al., 2023; Li et al., 2023b; Gu et al., 2024b), but this cannot be a fundamental solution.

Whether a dataset preserves or suppresses this multiplicity depends on design choices of multimodal pair collection and task definition: retaining only specific, narrowly defined examples may reduce some matching ambiguity, but this does not eliminate multiplicity and can be misaligned with general-purpose objectives. Bringing VL tasks as an example, we can reduce the potential matches of the given image by increasing specificity with long-form (Zhang et al., 2024) or all the localized details in the image (Pont-Tuset et al., 2020); this may reduce some spurious matches, but multiple unannotated long captions can still be valid for the same input, and the added detail is not always beneficial for general-purpose downstream tasks, such as zero-shot classification. On the other hand, if we focus on the salient objects in the image (Chen et al., 2015), the captioning process becomes cheaper, but the possible matching images per each caption will dramatically increase (Chun et al., 2022).

Lastly, recent dataset construction is increasingly automated via recaptioning or synthetic generation using (multimodal) large language models (Li et al., 2025). While this can improve scale and consistency, the generators themselves are not multiplicity-aware, often collapsing a set of valid alternatives into a single canonical description. When such synthetic pairs are further filtered by a fixed alignment scorer (*e.g.*, CLIP), a self-reinforced scorer-generator feedback loop can emerge: the scorer selects data that match its own

inductive biases, and the next generation step amplifies them. This loop risks cascading multiplicity-related failures by suppressing diverse but valid correspondences and reinforcing underspecified or stylistically narrow annotations.

### 5.2. Current attempts and future directions

Despite its importance, multiplicity has received limited attention in the context of dataset construction. While multiplicity-aware modeling and architecture design may eventually need to account multiplicity, minimizing unnecessary multiplicity at the dataset level remains a critical and cost-effective strategy, especially in the current paradigm where scaling-law still holds (Cherti et al., 2023).

Multiplicity should be considered even before data collection, *i.e.*, starting from task definition. Cross-modal alignment is inherently task-dependent. Previous works (Yu et al., 2023; Wu et al., 2024) showed that collecting task-relevant instances improves multimodal training. Without clear criteria for valid matches, datasets may introduce unintended multiplicity, causing downstream instability.

In addition, a careful multimodal pair collection process will be helpful to reduce the level of multiplicity. For example, filtering strategies should go beyond coarse alignment scores (*e.g.*, CLIP similarity) and explicitly target instances that amplify multiplicity (*e.g.*, underspecified inputs). One possible direction is a filtering based on specificity, such as HYPE (Kim et al., 2024b). By selecting more specific instances (defined by the embedding property), HYPE leads to higher-quality datasets and improved downstream performance. This supports the broader hypothesis that reducing multiplicity at the data level yields tangible benefits throughout the multimodal pipeline.

Finally, we can also consider explicitly multiplicity-aware data collection: instead of collecting paired instances, we can collect as many plausible counterparts as possible for each instance. For example, Zitnick & Parikh (2013) asked multiple participants to draw abstract images depicting the same sentence description, thereby collecting multiple semantically similar visual realizations of the same abstract description. This strategy preserves the many-to-many structure, but its scalability remains an open challenge, especially for large-scale, open-ended multimodal corpora.

## 6. Alternative Views

**Scaling under one-to-one supervision is sufficient.** Practitioners can argue that simply scaling models and datasets under sparse one-to-one annotations is enough to obtain strong multimodal systems (Radford et al., 2021; Jia et al., 2021; Cherti et al., 2023; Gadre et al., 2024). We agree that scaling can continue to improve average benchmark performance in the near term. However, our claim is that

this trend does not resolve multiplicity; it often *masks* it. As models become stronger, evaluation increasingly hinges on the fidelity of supervision and metrics: unobserved-but-valid correspondences create FNs and distort both training signals (Section 3) and retrieval-style evaluation (Section 4). This resembles how dataset imperfections become more consequential at high performance regimes in classification benchmarks (Beyer et al., 2020). These limitations are not purely hypothetical. For instance, Chun et al. (2022) shows that under sparse one-to-one annotations, many pairs treated as negatives are in fact valid matches for human annotators, revealing a larger set of positives than the original benchmark. Moreover, when evaluation accounts for multiple positives and precision metrics, the relative ranking of retrieval models and their correlation with human judgments can differ markedly from single-positive R@K evaluation. These observations support our claim that as models improve, benchmark reliability becomes increasingly limited by sparse supervision and metric choice under multiplicity.

Our view is that scale may reduce some symptoms of multiplicity, but it does not remove the underlying supervision mismatch. As observed by Sterz et al. (2025), even a strong MLLM, trained on large-scale data with many parameters (*e.g.*, GPT-4 (Achiam et al., 2023) or Gemini (Team et al., 2023)), still struggles when multiplicity is introduced in the benchmark. We believe that scaling may alleviate some effects in practice, but not truly address the problem.

In addition, as data pipelines increasingly rely on automated generation and filtering, the one-to-one paradigm risks reinforcing a narrow notion of "alignment" by scorer-generator feedback loops, potentially suppressing valid alternatives rather than capturing them. Thus, scaling may improve *scores*, while multiplicity remains a structural bottleneck for robustness, uncertainty, and human-aligned behavior.

**Multiplicity is an avoidable issue with good design.** A common alternative view is that multiplicity is not inherent: with cleaner data and better curation, it can be treated as noise; with well-defined, task-specific deployments it becomes practically irrelevant; and with more specific or long-form annotations the space of plausible matches shrinks enough that one-to-one supervision is "close enough". A good design can reduce *unnecessary* multiplicity. However, these arguments do not eliminate multiplicity. First, even under high-quality annotation, intra-modal variability, representational asymmetry, and task-dependent alignment imply that multiple correspondences can remain valid (Section 2). Second, practical deployments still face changing contexts and distribution shift, where latent multiplicity resurfaces as vulnerable evaluation or suppressed valid alternatives. Last, increasing specificity or constraining tasks reduces some spurious matches but introduces a specificity-generality trade-off: what is "correct" depends on the in-

tended notion of correspondence, and overly specific supervision can be misaligned with broad reuse (*e.g.*, zero-shot settings). Therefore, the key question is not whether multiplicity can be engineered away, but how to explicitly define correspondence and build training and evaluation protocols that remain faithful under a many-to-many structure. In other words, multiplicity can be *mitigated* by fixing a narrow notion of correspondence (e.g., higher specificity or tighter task constraints), but this inevitably trades off generality and does not eliminate the existence of multiple valid alignments in open-ended multimodal use.

## 7. Call to Action: A multiplicity-aware pipeline

Multiplicity is unlikely to be fully resolved generally, because its causes, such as modality asymmetry and task-dependent ambiguity, are structural properties of multimodal tasks rather than temporary artifacts or annotation noise. Our view is that the goal is not to recover perfect one-to-one correspondences (which is very difficult, as discussed in Appendix A), but to better account for multiplicity in practice. If multiplicity is not modeled directly, we still cannot tell which unobserved matches are valid, which negatives are actually false negatives, or when benchmark annotations are incomplete.

From this perspective, multiplicity can still be addressed to a meaningful extent: dataset construction can reduce underspecified pairs, training objectives can avoid treating all unobserved pairs as negatives, and evaluation can move from single-positive matching to multi-positive or ranking-sensitive protocols. These steps may not eliminate multiplicity, but they can substantially reduce the mismatch it creates. As we argue throughout the paper, multiplicity yields predictable failure modes under the one-to-one paradigm. We discuss these failure modes more in Appendix C.1. Now, we outline actionable directions that collectively sketch a multiplicity-aware pipeline.

Our main principle is simple: **even if multiplicity cannot be resolved perfectly, it should be taken into account when designing datasets, methods, and benchmarks.**

**Guidelines for practice.** (i) **Benchmark organizers:** for each query, publish a small *candidate pool* (*e.g.*, top-$M$ retrieved items from diverse baseline models), verify multiple positives, and report ranking-sensitive metrics. (ii) **Model builders:** report results under multiplicity-aware evaluation alongside standard one-to-one metrics. (iii) **Dataset builders:** explicitly filter *underspecified* instances (*e.g.*, overly generic captions) and release a "specificity" diagnostic so downstream users can control the trade-off between generality and matching ambiguity. (iv) **Reviewers/readers:** approach claims based on sparse one-to-one metrics with

caution when multiplicity is likely; look for evidence across precision-based metrics or verified multi-positive subsets.

**For modeling researchers.** (i) **Treat alignments as latent or set-valued:** develop objectives that consider multiple positives (or positive-unlabeled structure), mitigating FNs without assuming one-to-one. (ii) **Represent uncertainty and multiplicity:** explore multi- and distributional representations that can encode input and matching ambiguity. (iii) **Model beyond one-to-one mapping:** incorporate additional context to specify the intended correspondence, more discussions are in Appendix C.2.

**For benchmark designers.** (i) **Make relevance explicit:** benchmarks should specify the intended notion of correspondence, especially in task-dependent settings. (ii) **Move beyond single-positive:** whenever feasible, annotate or validate multiple positives and adopt ranking-sensitive metrics that reward multiple relevant matches (*e.g.*, mAP@R) rather than relying solely on R@K, which can be misleading under multiplicity. (iii) **Use human preference strategically:** human judgments provide a robust signal under semantic diversity and can be used to assist automatic metrics.

**For dataset builders.** (i) **Filter underspecified instances explicitly:** go beyond coarse alignment scores and target examples that amplify spurious correspondences (*e.g.*, overly generic captions). (ii) **Avoid scorer-generator collapse:** when using (M)LLM generation, reduce dependence on a single alignment scorer by using diverse scorers, holding out human audits, and incorporating diversity-aware constraints; otherwise, feedback loops can progressively suppress valid alternatives. (iii) **Embrace task-conditioned data views:** when a dataset is intended for broad reuse, consider releasing multiple task-conditioned subsets or annotation views that reflect different correspondence notions, rather than forcing a single global alignment definition.

**Across the pipeline.** **Iterative development cycles:** treat dataset collection, filtering, modeling, and evaluation as a coupled loop, under multiplicity-aware frameworks. As shown in the unimodal dataset construction (Benenson et al., 2019), such iteration will significantly improve dataset quality and system robustness over time.

## Acknowledgements

This work is supported by the Princeton Francis Robbins Upton Fellowship to S.C. We are grateful to Esin Tureci, Tyler Zhu, Junsuk Choe, and Song Park for valuable feedback that helped shape this work.

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

# Appendix

## A. Unimodal Classification vs. Multimodal Learning

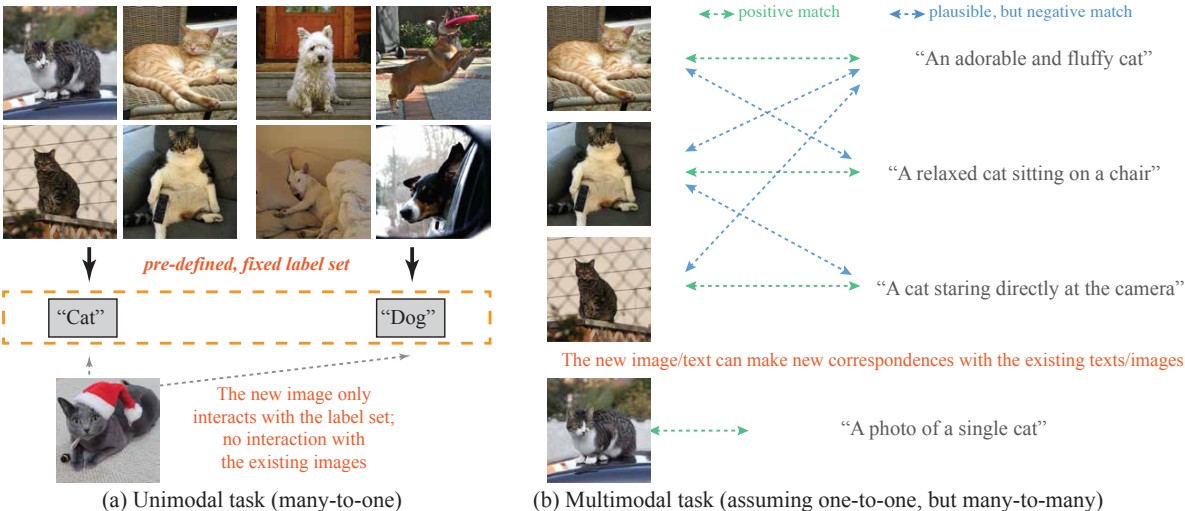

(a) Unimodal task (many-to-one)    (b) Multimodal task (assuming one-to-one, but many-to-many)

*Figure A.1.* **How unimodal and multimodal tasks are different?** Classification tasks assume a fixed label set. Even though we add more instances in the dataset, the number of correspondences increases constantly, and the new instance does not affect to the existing instances. However, the correspondences in multimodal datasets, assuming one-to-one mapping, increase $O(N)$ by adding one multimodal pair.

In this section, we provide an overview of the difference between the unimodal classification and the multimodal data pipelines. We will show that the multimodal dataset construction pipeline is fundamentally brittle to the multiplicity problem, while a carefully designed unimodal dataset pipeline can often suppress the effect of multiplicity.

In unimodal classification, $\mathcal{Y}$ is a fixed and pre-defined label set (*e.g.*, class labels), *i.e.*, $|\mathcal{Y}|$ is constant. Furthermore, unimodal label sets are usually well-defined; there are few cases where $x \in \mathcal{X}$ belongs to multiple $y \in \mathcal{Y}$. Although some studies argued that popular classification benchmarks can be viewed as multi-labeled (Beyer et al., 2020; Shankar et al., 2020; Yun et al., 2021), the degree of multiplicity is relatively limited compared to multimodal tasks. For example, Yun et al. (2021) showed that while ImageNet images may correspond to multiple valid labels, roughly five labels per image can account for most of the semantic ambiguity. Hence, adding a new instance $x$ does not change "ground-truths" of existing data points, since labels come from a fixed set (See Figure A.1 (a)).

In contrast, multimodal datasets define ground truth through cross-modal relations and are typically collected as a sparse set of annotated positive pairs $(x, y)$, under an one-to-one correspondence assumption. Since multimodal tasks rely on *pairwise matching*, adding a new pair can create $O(N)$ additional *plausible* correspondences with existing instances, even though only a tiny subset is annotated. Moreover, unlike unimodal label sets that can be curated to reduce

overlap (*e.g.*, via WordNet hierarchies (Russakovsky et al., 2015) or balanced popularity (Kuznetsova et al., 2020)), the nature of multimodal data collection is highly diverse, introducing multiple sources of multiplicity. Consequently, new annotations can induce additional implicit matches with many existing instances. For example, adding a caption like "a photo" introduces plausible matches not just with one image, but potentially with many photographic images in the dataset (see Figure A.1 (b)). Thus, the set of plausible correspondences can expand rapidly as the dataset scales. In practice, many instances have multiple valid counterparts in the other modality, *i.e.*, $|\{y \in \mathcal{Y} : (x, y) \in \mathcal{R}\}| > 1$ for some $x$. This contrast makes multiplicity particularly problematic in multimodal learning: scaling the dataset changes not only the number of examples, but also the space of plausible cross-modal relations to be learned and evaluated.

## B. Human preference vs. evaluation metrics

Previous studies (Musgrave et al., 2020; Chun et al., 2022) have shown that R@K is not only less informative than ranking-based metrics such as mAP@R (where $R$ denotes the number of positives), but can also be misleading. In particular, R@K ignores the overall ranking quality and fails to reward models that retrieve multiple semantically appropriate items, making it insensitive to models that produce coherent and diverse outputs; it makes a case when R@K is 100% but mAP@R is not 100% (See Figure B.1 (B)).

Query caption: "A train on a train track near many trees"

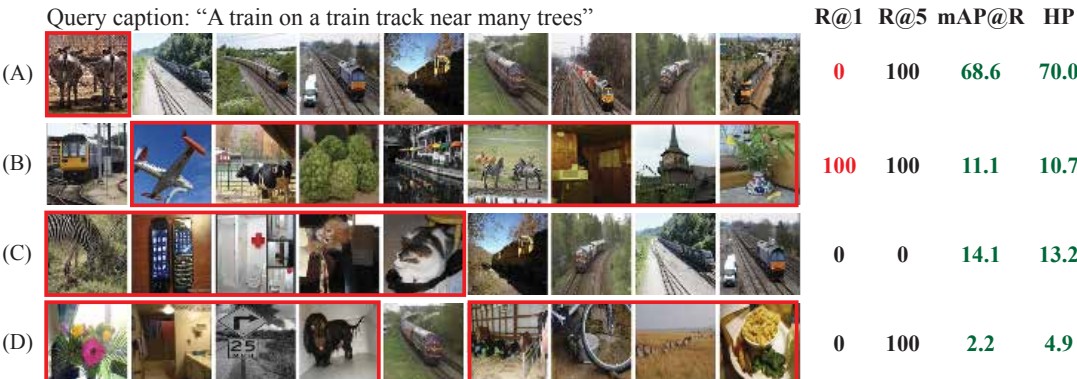

*Figure B.1.* **Human preference vs. evaluation metrics under multiplicity.** Chun et al. (2022) asked human annotators to compare four retrieval scenarios: (A) only top-1 is wrong, (B) only top-1 is correct, (C) top-1 to top-5 are wrong, and (D) only top-5 is correct. mAP@R (Musgrave et al., 2020) is highly correlated to human preference (HP), while R@Ks are often irrelevant.

We borrow the experimental result and the figure from Chun et al. (2022). Figure B.1 shows the overview of a human preference study with four different retrieval scenarios. Given a text query (*e.g.*, "A train on a train track near many trees"), assume there are four retrieval systems that return top-k similar items in four different precisions: (A) only the top-1 item is wrong, while the other items are correct, (B) only the top-1 item is correct, but the others are wrong, (C) items from top-1 to top-5 are wrong, but the others are correct, (D) only the top-5 item is correct. In Figure B.1, R@1, R@5, and mAP@R scores of each system are shown. For example, System A shows 0% R@1, but the best mAP@R among the systems; while System B shows 100% R@1 and R@5, while its mAP@R is significantly lower then system A. Chun et al. (2022) asked human annotators to choose a more preferable system by pairwise comparison. After the pairwise comparison, they reconstruct the underlying human preference by the linear BT model (Bradley & Terry, 1952). Interestingly, the human preference (HP) score is highly aligned with mAP@R, while R@K cannot capture the human preference. Unfortunately, enlarging $K$ cannot be a solution; Chun et al. (2022) showed that the rankings by R@K with different $K$s are highly correlated with each other, while the ranking by mAP@R is less correlated with them. This indicates the need for carefully annotated cross-modal retrieval benchmarks and more reliable evaluation metrics for retrieval benchmarks under multiplicity.

This suggests that evaluation under multiplicity should verify multiple positives whenever feasible and report ranking-sensitive metrics in addition to (or instead of) R@K.

## C. More discussions

### C.1. Predictable failure modes of one-to-one supervision

Throughout the paper, we argue that multiplicity yields *predictable failure modes* under the one-to-one paradigm: (i)

as datasets and models scale, the rate of unobserved-but-valid correspondences grows, increasing training instability and degrading representation quality; and (ii) evaluation benchmarks with sparse annotations become increasingly unreliable. These predictions are empirically testable by varying dataset scale and annotation density, and they motivate concrete changes to how we design training, evaluation, and dataset construction pipelines.

We provide suggestive empirical evidence for these predictions. For example, Chun et al. (2022) has shown that re-annotating unobserved-but-valid correspondences as positives and using precision-based metrics can substantially change the ranking of models. Specifically, models with high R@K scores (which focus on exact matching) often degrade under precision-based metrics such as mAP@R. Similarly, the authors show that even at the relatively small scale of image-text pairs in COCO Caption, the number of hidden false negatives is nontrivial.

### C.2. More discussions on novel modeling

We suggest to incorporate additional context to specify the intended correspondence (*e.g.*, text-conditioned transformations (Gu et al., 2024a), spatial grounding via local regions/masks (Lee et al., 2024; Cai et al., 2024), or lexically specifying the characteristic of the corresponding audio from the video (Jeong et al., 2025)), and explore compositional representations that model an instance as a composition of underlying concepts (Ma et al., 2023).

As another example, distribution-aware (or multi-prompt) adaptation methods (Wang et al., 2023; Cho et al., 2023; Liu et al., 2024) are relevant as evidence that downstream VLM adaptation often benefits from representing a class or concept with multiple prompts or distributions, which is consistent with our broader claim that multimodal semantics are often non-unique.

Also, we suggest extending the multiplicity-aware pipeline to generative VLMs: one-reference supervision can suppress valid alternatives; future work should investigate multi-reference protocols, preference-based objectives, or distribution-aware training signals that better reflect the space of valid outputs.

### C.3. Relation to existing work

Multiplicity is closely related to several phenomena previously studied in multimodal learning, including modality asymmetry, ambiguity, information imbalance, and modality gap. These phenomena are important in their own right, and prior work has provided valuable evidence that multimodal representations can be affected by the mismatch between modalities. Our perspective is complementary: we view these phenomena as distinct mechanisms that can induce the same downstream structural problem, namely that an observed one-to-one pair often under-specifies the full set of valid cross-modal correspondences.

This distinction is important because mitigating one source of mismatch does not necessarily eliminate multiplicity. For example, reducing intra-modal variability or partially mitigating modality asymmetry may still leave multiple plausible correspondences due to task-dependent notions of relevance. Similarly, even if a benchmark is carefully curated for a particular notion of alignment, hidden valid matches can remain whenever relevance is many-to-many. Thus, multiplicity is not simply a collection of isolated issues, but a structural property of multimodal correspondence that affects data construction, training, and evaluation.

This perspective also clarifies the relation between multiplicity and modality gap studies (Liang et al., 2022; Levi & Gilboa, 2025; Schrodi et al., 2025). These works primarily analyze representational mismatch between modalities, especially in CLIP-style contrastive vision-language models. Such analyses are highly relevant to our argument because the modality gap and information imbalance can be viewed as representation-level manifestations of modality asymmetry, one of the sources of multiplicity. However, multiplicity is broader than the modality gap alone. While modality gap concerns the geometry of learned embeddings, multiplicity concerns the underlying correspondence structure: before any model is trained, a single image, caption, audio clip, or instruction may already admit multiple valid counterparts depending on semantic overlap, modality asymmetry, and task context.

Consequently, solutions that reduce the modality gap do not necessarily resolve the many-to-many nature of multimodal correspondence. Existing modality-gap analyses typically remain within a one-to-one training or evaluation paradigm, whereas our position is that this paradigm itself becomes insufficient when multiple plausible correspondences exist.

Moreover, modality gap studies mainly focus on CLIP-style training, while multiplicity affects a broader multimodal pipeline, including dataset construction, contrastive and generative training, and evaluation protocols. Our contribution is therefore to unify these related phenomena under a single structural view and to argue that multiplicity should be treated as a first-class consideration throughout the multimodal learning pipeline.

