# OpenReview forum: "Position: Multiplicity is an Inevitable and Inherent Challenge in Multimodal Learning"
_ICML.cc/2026/Position_Paper_Track — ICML 2026 Position Paper Track regular_

### Official Review · Reviewer_NoQD · 2026-03-12

**Significance:** 3
**Argument Clarity:** 3
**Rating:** 5
**Confidence:** 4

**Questions:**

1. Multiplicity is truly an important point for multimodal learning. However, recent works attempt to mitigate it by pairing images with more concrete, distinguishable captions or by introducing diverse tasks to achieve implicit alignment. Do these works already inherently consider multiplicity?
2. In my view, the challenge of multiplicity arises from the difficulty of assigning an instance a unique paired multimodal representation. Due to the asymmetry between modalities (as you mentioned), it seems this cannot be fully resolved. To what extent can this challenge actually be addressed?
3. Can you explain why one-to-one annotations remain sparse even with the increasing scale of models and datasets discussed in Section 6.1?
4. The paper focuses mainly on the contrastive learning paradigm, highlighting explicit alignment introduced by human annotations. Given the scaling of data, models, and tasks, can we alleviate or even mitigate the multiplicity challenge without explicitly considering the multiplicity?

**Alternative Views Section:**

Yes

**Compliance With Llm Reviewing Policy A Conservative:**

Affirmed.

**Discussion Potential:**

3

**Final Justification:**

My concerns are addressed, thus I keep the posive score.

**Paper Summary:**

This paper articulates multiplicity as a structural, rather than incidental, challenge in multimodal learning. It is well-connected to recent practices and addresses all three stages of the pipeline, i.e., data, training, and evaluation.

**Position:**

Yes

**Position In Title:**

Yes

**Related Work:**

3

**Strengths And Weaknesses:**

Strengths:
1. The definition of multiplicity is clearly established, making the paper well-argued and easy to follow.
2. The argument is comprehensive, covering various stages including data construction, training, and evaluation.
3. The investigation into multiplicity is intriguing and in-depth. It highlights a key factor that should be actively considered rather than overlooked.

Weaknesses:
1. The claim regarding the `sparse sampling of the potential correspondence space` is unclear. An explicit discussion should be included to clarify this concept.
2. Regarding the training stage, the paper highlights training objectives and the significance of negatives. However, it should also consider model architectures as an additional section to analyze how they influence multiplicity. This is a perspective that is currently under-explored in the main body.
3. The case presented in the introduction is intuitive but not entirely representative of recent Large Multimodal Models (LMMs). Multimodal learning tasks are diverse. Beyond explicit alignment, much recent work emphasizes implicit alignment to avoid imposing strong priors or relying solely on labeled paired data.

**Support:**

3

---

> ### Author Rebuttal · Authors · 2026-03-31
>
> Thank you for your encouragement and positive feedback. We address your comments below and will revise the paper accordingly.
>
>
> **[W1, Q3] Why do one-to-one annotations remain sparse?**
>
> We described the details of this in Appendix A. The main idea is that whenever we add a new multimodal pair (e.g., an aligned image-text pair) to the dataset, the number of possible annotations (e.g., matched or unmatched) is increased by O(N), where N is the size of the existing dataset. On the other hand, for unimodal tasks (such as classification), adding a new instance does not increase the order of new annotations. It is always O(K) where K is the number of classes (i.e., constant).
>
> Therefore, in practice, if we have a very large-scale multimodal paired dataset with 1 billion pairs, the number of possible annotations is quadratic in the dataset size; in this case, $10^{18}$. It is impossible to annotate all the possible cross-modal correspondences, which makes the annotations remain sparse.
>
> **[W2] Model architectures**
>
> Thanks for your suggestion. Our paper already discussed the architecture changes. Multi-head embeddings (L233-248) and probabilistic embeddings (L249-259) have different architectures from standard deterministic one-to-one mapping models. Multi-head embeddings produce multiple candidate embeddings, while probabilistic embeddings represent uncertainty through distributional parameters such as mean and variance. We will emphasize that there are several attempts to solve this problem using novel model architectures.
>
> **[W3, Q1] Not entirely representative of LMMs / Recent works attempt to mitigate it**
>
> Thank you for raising this point. We agree that the current manuscript focuses more on non-LLM-based VLM settings than on recent LMMs.
>
> We believe the same concern extends to multimodal generative models, although the concrete failure modes differ. For example, if training focuses on only one observed target even when multiple targets are valid, the model is still pushed toward an artificially narrow conditional distribution (as we briefly discuss in Sec. 3.1).
>
> In generative or instruction-tuned LMMs, training on a single observed response can suppress other valid responses and reduce calibrated diversity. Recent work, such as DARE [1], also suggests that strong LMMs (e.g., GPT4, Gemini) still struggle once evaluation explicitly allows multiple correct answers, i.e., when multiplicity is introduced in the benchmark. We will strengthen this discussion in the revision.
>
> - [1] DARE: Diverse Visual Question Answering with Robustness Evaluation
>
> **[Q2] To what extent can this challenge actually be addressed?**
>
> We agree that this challenge is unlikely to be fully resolved in a general sense, because the causes of multiplicity, such as modality asymmetry and task-dependent ambiguity, are structural properties of multimodal tasks rather than temporary artifacts or annotation noise. Our view is that the goal is not to recover perfect one-to-one correspondences (which is very difficult as discussed in Appendix A), but to better account for multiplicity in practice.
>
> From this perspective, multiplicity can still be addressed to a meaningful extent: dataset construction can reduce underspecified pairs, training objectives can avoid treating all unobserved pairs as negatives, and evaluation can move from single-positive matching to multi-positive or ranking-sensitive protocols. These steps may not eliminate multiplicity, but they can substantially reduce the mismatch it creates.
>
> **[Q4] Can we mitigate the multiplicity challenge without explicitly considering the multiplicity?**
>
> We believe recent trends, such as scaling data, models, and tasks, can mitigate some of the practical effects of multiplicity, even without modeling it explicitly. Better captions, broader supervision, and stronger implicit alignment may reduce ambiguity in some settings.
>
> But that does not make the problem go away. If multiplicity is not modeled directly, we still cannot tell which unobserved matches are valid, which negatives are actually false negatives, or when benchmark annotations are incomplete. Therefore, while scaling may soften some symptoms in practice, it does not address the source of the mismatch. That is why we think multiplicity should be treated as a first-class issue, not something scaling will absorb.

---

> > ### Author Rebuttal · Reviewer_NoQD · 2026-04-03
> >
> > Thanks for the rebuttal. My concerns are addressed. I will keep the posive score.

---

> > > ### Author Response · Authors · 2026-04-04
> > >
> > > We thank you again for your positive feedback and for confirming that your concerns have been fully addressed. If there are any other aspects that you think could further improve the clarity, positioning, or impact of the paper for the final version (or any future work), we would greatly appreciate any additional suggestions.

---

### Official Review · Reviewer_XyCk · 2026-03-12

**Significance:** 3
**Argument Clarity:** 2
**Rating:** 4
**Confidence:** 4

**Questions:**

1、The paper groups intra-modal variability, modality asymmetry, and task-dependent alignment under a single umbrella. Could the authors clarify why these should be treated as one coherent challenge, rather than several related but conceptually distinct issues that may require different forms of analysis and intervention?

2、The paper suggests that the standard one-to-one assumption is fundamentally mismatched with real-world multimodal data. However, as model and dataset scales continue to increase, is it possible that large models can implicitly absorb or mitigate some effects of multiplicity? If so, why should multiplicity still be viewed as a core bottleneck rather than a secondary imperfection?

3、Much of the paper’s reasoning is grounded in retrieval-style settings. Could the authors clarify whether they believe the same position applies equally strongly to generative multimodal models, instruction-tuned VLMs, and multimodal reasoning systems, and if so, through what concrete failure modes?

**Alternative Views Section:**

Yes

**Compliance With Llm Reviewing Policy A Conservative:**

Affirmed.

**Discussion Potential:**

3

**Final Justification:**

The paper presents a novel and promising idea and its contributions are sufficient to merit a borderline accept at ICML.

**Paper Summary:**

This position paper argues that multiplicity is an inevitable and inherent challenge in multimodal learning. Addressing core challenges such as intra-modal variability, asymmetry across modalities, and task-dependent alignment, the authors demonstrate how the common one-to-one assumption can distort training, evaluation, and dataset construction in multimodal systems. Furthermore, the paper candidly examines opposing viewpoints, acknowledging that scaling and better data design may alleviate some practical issues while arguing that these strategies do not eliminate the underlying many-to-many nature of real-world multimodal correspondence.

**Position:**

Yes

**Position In Title:**

Yes

**Related Work:**

2

**Strengths And Weaknesses:**

Strengths
1. The paper highlights multiplicity as a fundamental challenge in multimodal learning, offering a refreshing perspective for the field.
2. The paper proposes that multimodal correspondence is inherently many-to-many rather than one-to-one, and systematically analyzes how this affects dataset construction, model training, and evaluation.
3. Regarding counterarguments, the authors explicitly discuss alternative views, such as the sufficiency of scaling and better data design, and provide reasoned responses while acknowledging their practical appeal.

Weaknesses
1、The authors argue that multiplicity introduces a large number of false negatives and fundamentally distorts training and evaluation. However, this claim is supported primarily through conceptual reasoning and selective references to prior work, rather than direct evidence showing how prevalent or severe the issue is in modern large-scale multimodal datasets. As a result, the practical magnitude of the problem remains unclear.

2、The proposed framework groups together intra-modal variability, modality asymmetry, and task-dependent alignment. While this unification is intuitively appealing, these issues arise from different sources and may require different conceptual treatments. The paper does not sufficiently justify why they should be treated as one coherent problem rather than a collection of related but distinct challenges.

3、Much of the paper’s reasoning about false negatives, matching ambiguity, and metric failure is grounded in retrieval-style settings. Although the authors briefly mention generative models and broader multimodal systems, the discussion remains limited. This makes it difficult to assess whether the claimed position extends with equal force to generation, reasoning, or decision-making tasks.

4、The paper acknowledges that scale and improved dataset design may alleviate some practical issues, but its responses remain largely suggestive. In particular, the possibility that large-scale models may implicitly absorb or mitigate some effects of multiplicity is not examined in depth. As a result, the counterarguments are raised, but not decisively resolved.

5、There are minor writing issues. For example, the paper contains a typo (in line 228, “Nonethelese” should be “Nonetheless”), which slightly affects presentation quality.

**Support:**

2

---

> ### Author Rebuttal · Authors · 2026-03-31
>
> We deeply appreciate your thoughtful and encouraging review. We address all the raised concerns by the reviewers below and will revise the paper accordingly.
>
> **[W1] Missing direct evidence**
>
> Thanks for pointing this out. We would like to emphasize that our main point is that multiplicity is inherent and inevitable in multimodal learning and affects the full pipeline.
>
> In the paper, we already discussed several empirical observations from prior work that show the existence of this problem. We briefly restate several empirical findings from prior work here for clarification. In the revision, we will make this evidence more explicit in Sec 3.1, 4.1, and 5.1.
>
> For evaluation, [1] empirically shows that a popular image-caption dataset (COCO Caption) misses many valid matches: each caption is associated with substantially more valid images than the original benchmark records (e.g., originally 1 valid image, but human-verified positives were 8.5 for each caption). The paper also shows that re-evaluating VLMs with corrected associations can noticeably change model rankings (Kendall’s tau is 0.47). This is direct evidence that one-to-one benchmarks can become unreliable under multiplicity.
>
> - [1] ECCV Caption: Correcting false negatives by collecting machine-and-human-verified image-caption associations for MS-COCO
>
> [2] provides a relevant example on the dataset side. [2] proposes a new dataset selection algorithm based on “specificity”, which measures the degree of semantic overlap between data points. By filtering out underspecified image-text pairs, [2] improves training data quality. In other words, a dataset with a lower degree of multiplicity (in [2], they stated it as “specificity”) leads to a better trained model.
>
> - [2] HYPE: Hyperbolic Entailment Filtering for Underspecified Images and Texts.
>
> On the training side, [3] shows a related failure case. [3] showed that the existing training methods assuming a strong one-to-one correspondence (by hardest negative mining) can break down as model scale increases (achieving 40.0 on ViT-B/32 but 20.2 on ViT-L/14), while approaches that account for multiplicity remain much more stable (40.1 on ViT-B/32, 42.1 on ViT-L/14). In summary, [3] shows that if the effect of false negatives (due to multiplicity) becomes more significant, conventional training strategies can fail.
>
> - [3] Improved Probabilistic Image-Text Representations
>
>
> **[W2, Q1] Why should different problems be treated as one problem, multiplicity**
>
> We do not claim that these issues are identical. Our claim is that they are distinct sources of the same downstream challenge: the observed one-to-one pair often under-specifies the full space of valid cross-modal correspondences. Each issue is important in its own right, but addressing only one of them does not remove multiplicity, because the remaining sources can still produce multiple plausible alignments.
>
> This is why we treat multiplicity as a coherent challenge at the level of data, training, and evaluation. For example, in a retrieval benchmark, the practical problem is the existence of hidden plausible matches that make one-to-one annotations unreliable [1]. Reducing intra-modal variability alone, or partially mitigating modality asymmetry, does not guarantee that these hidden matches disappear. In that sense, the individual issues matter, but multiplicity remains the downstream problem that must be addressed directly.
>
>
> **[W3, Q3] Does the same position apply equally strongly to generative multimodal models?**
>
> We think that the same position can be applied almost equally strongly to multimodal generative models. If training focuses on only one observed target even when multiple targets are valid, the model is still pushed toward an artificially narrow conditional distribution (as we briefly discuss in Sec. 3.1).
>
> This matters beyond retrieval. In generative or instruction-tuned multi-modal large language models (MLLMs), training on a single observed response can suppress other valid responses and reduce calibrated diversity. Recent work, such as DARE [4], also suggests that strong MLLMs (e.g., GPT4, Gemini) still struggle once evaluation explicitly allows multiple correct answers, i.e., when multiplicity is introduced in the benchmark. We will strengthen this discussion in the revision.
>
> - [4] DARE: Diverse Visual Question Answering with Robustness Evaluation
>
> **[W4, Q2] Scaling laws**
>
> Our view is that scale may reduce some symptoms of multiplicity, but it does not remove the underlying supervision mismatch. As we mentioned in [W3, Q3], even a strong MLLM, trained on large-scale data with many parameters, still struggles when multiplicity is introduced in the benchmark. We believe that scaling may alleviate some effects in practice, but not truly address the problem.
>
> **[W5] Typos**
>
> Thanks. We will fix it and will check other grammar errors.

---

> > ### Author Rebuttal · Reviewer_XyCk · 2026-04-03
> >
> > Thank you for the author's reply. I will keep the original score.

---

> > > ### Author Response · Authors · 2026-04-04
> > >
> > > We thank you again for your positive feedback and for confirming that our response has fully addressed your concerns.
> > >
> > > Given that the main concerns on empirical grounding, the unifying perspective on multiplicity, and the extension beyond retrieval settings have been clarified, we would greatly appreciate any further guidance on what aspects may still limit the overall strength or conviction of the position above the borderline.

---

### Official Review · Reviewer_FFsh · 2026-03-16

**Significance:** 3
**Argument Clarity:** 3
**Rating:** 5
**Confidence:** 4

**Questions:**

1. There exists a domain of work on multi-label classification [X]. Similarly, there are datasets for multimodal  multi-label classification [Y]. Can the authors discuss and elaborate on how to use these existing multi-label datasets for developing novel models considering multiplicity?

[X] Tarekegn et al., Deep Learning for Multi-Label Learning: A Comprehensive Survey, 2024.

[Y] Mangolin et al., A multimodal approach for multi-label movie genre classification, 2020.

**Alternative Views Section:**

Yes

**Compliance With Llm Reviewing Policy A Conservative:**

Affirmed.

**Discussion Potential:**

3

**Final Justification:**

The rebuttal has clarified my major concerns. I have improved my rating to 5 after the rebuttal. I am positive about the paper's contributions.

**Paper Summary:**

The paper argues that multiplicity is an inevitable and inherent challenge in multimodal learning which affects the entire pipeline - data construction, contrastive training and the evaluation phase. The paper formalizes the causes and consequences of the many-to-many relationships between data from different modalities. The paper argues that multiplicity creates ambiguity when multi-modal models are trained contrastively assuming one-to-one multi-modal pairs of data. The authors propose to build datasets, model training approaches and evaluations which explicitly considers the multiplicity nature of multimodal data.

**Position:**

Yes

**Position In Title:**

Yes

**Related Work:**

3

**Strengths And Weaknesses:**

Strengths -
1. The paper focus on the many-to-many nature of multi-modal data and discusses limitations of multi-modal models trained with contrastive learning objectives.
2. The paper discusses existing works and provides arguments on how to move towards considering multiplicity in dataset construction, training approaches and evaluation practices.

Weaknesses-
1. A major concern is that this is not a new position. Existing works [a,b] have analyzed multiplicity in multi-modal learning and stated similar views on multiplicity or asymmetry between modalities (information imbalance) in recent works.
2. While the paper propose to move beyond one-to-one mapping, there is no specific proposal for that beyond discussion of existing efforts in that direction.
3. The paper lacks discussion on existing multi-label multi-modal datasets.
4. The paper does not provide a proper conclusion section to conclude the arguments.

[a] Two effects, one trigger: On the modality gap, object bias, and information imbalance in contrastive vision-language models. ICLR 2025.

[b] The double-ellipsoid geometry of clip. ICML 2025.

**Support:**

3

---

> ### Author Rebuttal · Authors · 2026-03-31
>
> We thank the reviewers for constructive feedback. We address all the raised concerns by the reviewers and will update our paper accordingly.
>
> **[W1] Not a new position**
>
> We believe that our position is a new position, although there exist related prior works. Our main argument is that **“multiplicity is an inherent and inevitable problem in multimodal learning, and it affects the entire multimodal learning pipeline”**. Our claim is not that prior work has never discussed asymmetry, ambiguity, modality gap, or information imbalance in multimodal learning. Rather, our contribution is to unify and elevate these separate phenomena into a single structural property of multimodal correspondence (i.e., multiplicity), and to show how this structural property can affect the entire multimodal pipeline.
>
> Our claim is broader than [a, b]. [a, b] mainly focus on the asymmetry between modalities, more specifically, the modality gap. Our point is not merely that modalities are asymmetric, but that this asymmetry is only one source of a broader many-to-many correspondence structure (i.e., multiplicity) which affects the entire multimodal pipeline. [a, b] studied the modality gap problem without considering the existence of multiple plausible answers (i.e., multiplicity). Their solutions still rely on the one-to-one mapping assumption, whereas our argument is that the one-to-one mapping framework is insufficient under multiplicity. Also, they only focus on CLIP training, while we consider broader and more general multimodal learning scenarios.
>
> **[W2] No specific proposal**
>
> Our paper already contains several actionable items throughout the paper (Sec 3.2, 4.2, 5.2, and 6.2). Specifically, we already listed several concrete directions, including:
>
> - (i) moving from single-positive to multi-positive / ranking-sensitive evaluation,
> - (ii) treating alignments as latent or set-valued rather than strictly one-to-one,
> - (iii) developing representations that capture uncertainty and multiplicity, and
> - (iv) filtering underspecified instances during dataset construction.
>
> Especially for “moving beyond one-to-one mapping”, we already provided a specific proposal in (ii): we should treat multimodal alignments as latent or set-valued rather than deterministic one-to-one correspondences.
>
> **[W3, Q1] Multi-label multi-modal datasets**
>
> We agree that multi-label classification is partially relevant to our scenario, especially because it also moves beyond single-target supervision. However, our setting differs in an important way: multi-label classification assumes a fixed label set, whereas multimodal multiplicity arises from open-ended many-to-many correspondences across instances and modalities. We already discussed this in Appendix A.
>
> Our paper focuses on a multimodal learning scenario, where [X] is for a unimodal classification problem. One particular problem in our case is that it is almost impossible to exhaustively annotate all plausible correspondences. As discussed in Sec A (Appendix), the annotation cost of unimodal tasks is not affected by a new instance (because the label set is fixed), but that of multimodal tasks is highly affected by a new instance, because a new instance can be plausibly matched with the existing one in the dataset. As far as we understand, [Y] also focuses on a classification task. Our setting is not a fixed-label classification setting; instead, it focuses on cross-modal relevance/alignment between instances. There is no specific label set, but we only use the alignment or relevance between instances of each modality. Therefore, our scenario is different from [X,Y] or multi-label datasets.
>
> **[W4] No conclusion section**
>
> Section 6.2 already concludes the main argument and proposes actionable items. We believe Section 6.2 serves as a valid conclusion section. We are willing to rename Section 6.2 as an explicit conclusion section.

---

> > ### Author Rebuttal · Reviewer_FFsh · 2026-04-03
> >
> > I thank the authors for the response. The authors clarify and address most of my concerns. I would like the authors to add a detailed discussion of information imbalance and modality gap from the papers [a,b] in the revised version of the paper. These papers I believe serve as the basis for multiplicity in multimodal models. I will increase my score.

---

> > > ### Author Response · Authors · 2026-04-04
> > >
> > > We are glad that our responses helped resolve your concerns. Thank you again for your thoughtful feedback – absolutely, we will certainly add a discussion of [a,b] in the next version. If there are any other aspects that you think could further improve the clarity, positioning, or impact of the paper for the final version (or any future work), we would greatly appreciate any additional suggestions.

---

### Official Review · Reviewer_sQLR · 2026-03-19

**Significance:** 3
**Argument Clarity:** 3
**Rating:** 5
**Confidence:** 4

**Questions:**

1), It is unlikely that a “perfect” embedding can ever be achieved. Similarly, it is impractical to construct a “perfect” multimodal dataset during data collection. Therefore, the paper should discuss the fundamental principles or guiding strategies for mitigating the impact of multiplicity in practice.

2), missing references about distributed learning [1,2,3].




[1] Liu et.al., Patch-Prompt Aligned Bayesian Prompt Tuning for Vision-Language Models.

[2] Cho et.al., Distribution-Aware Prompt Tuning for Vision-Language Models.

[3] Wang et.al., Tuning Multi-mode Token-level Prompt Alignment across Modalities.

**Alternative Views Section:**

Yes

**Compliance With Llm Reviewing Policy A Conservative:**

Affirmed.

**Discussion Potential:**

4

**Final Justification:**

The rebuttal has addressed most of my concerns, and I keep my original score as 5.

**Paper Summary:**

This position paper challenges the one-to-one alignment paradigm in current vision-language models (VLMs) and introduces the concept of multiplicity (i.e., many-to-many alignment) to better capture inherent uncertainty. The core argument is that existing single-vector embedding approaches fail to represent the full semantic richness of features, particularly in multimodal settings. Specifically, the authors analyze the issue of multiplicity from the perspectives of datasets, training objectives, and evaluation benchmarks, arguing that neglecting multiplicity can harm the quality of learned representations in VLMs. Finally, the paper provides a detailed discussion of the relationship between multiplicity and recent advances, and proposes several promising future directions for VLM research.

**Position:**

Yes

**Position In Title:**

Yes

**Related Work:**

2

**Strengths And Weaknesses:**

Strengths:

1), The motivation of this position paper is interesting and thought-provoking. Uncertainty is a fundamental aspect of the real world, yet it has received limited attention in current VLMs. This paper highlights the concept of multiplicity across datasets, training objectives, and evaluation benchmarks, helping readers better understand its potential role in VLMs.

2), Detailed discussions are provided to explain the relationship between multiplicity and recent attempts, which not only explain the multiplicity in another perspective but also point out some valuable research directions.

Weaknesses:

1), Empricial findings are missing to prove the multiplicity issue in dataset, training and benchmarks.

2), Given the fact that more powerfull VLMs represent more accurent meaning and reasoning, this paper should discuss more about the scaling law in dataset and model parametes and the multiplicity issue.

**Support:**

3

---

> ### Author Rebuttal · Authors · 2026-03-31
>
> Thank you for the thoughtful and encouraging review. We are glad the paper’s main position came across clearly. We address your comments below and will revise the paper accordingly.
>
> **[W1] Empirical findings are missing**
>
> Thanks for pointing this out. We would like to emphasize that our main point is that multiplicity is inherent and inevitable in multimodal learning and affects the full pipeline.
>
> In the paper, we already discussed several empirical observations from prior work that show the existence of this problem. We briefly restate several empirical findings from prior work here for clarification. In the revision, we will make this evidence more explicit in Sec 3.1, 4.1, and 5.1.
>
> For evaluation, [1] empirically shows that a popular image-caption dataset (COCO Caption) misses many valid matches: each caption is associated with substantially more valid images than the original benchmark records (e.g., originally 1 valid image, but human-verified positives were 8.5 for each caption). The paper also shows that re-evaluating VLMs with corrected associations can noticeably change model rankings (Kendall’s tau is 0.47). This is direct evidence that one-to-one benchmarks can become unreliable under multiplicity.
>
> - [1] ECCV Caption: Correcting false negatives by collecting machine-and-human-verified image-caption associations for MS-COCO
>
> [2] provides a relevant example on the dataset side. [2] proposes a new dataset selection algorithm based on “specificity”, which measures the degree of semantic overlap between data points. By filtering out underspecified image-text pairs, [2] improves training data quality. In other words, a dataset with a lower degree of multiplicity (in [2], they stated it as “specificity”) leads to a better trained model.
>
> - [2] HYPE: Hyperbolic Entailment Filtering for Underspecified Images and Texts.
>
> On the training side, [3] shows a related failure case. [3] showed that the existing training methods assuming a strong one-to-one correspondence (by hardest negative mining) can break down as model scale increases (achieving 40.0 on ViT-B/32 but 20.2 on ViT-L/14), while approaches that account for multiplicity remain much more stable (40.1 on ViT-B/32, 42.1 on ViT-L/14). In summary, [3] shows that if the effect of false negatives (due to multiplicity) becomes more significant, conventional training strategies can fail.
>
> - [3] Improved Probabilistic Image-Text Representations
>
> **[W2] Powerful VLMs (Scaling law) vs. Multiplicity**
>
> Our view is that scale may reduce some symptoms of multiplicity, but it does not remove the underlying supervision mismatch. If training focuses on only one observed target even when multiple targets are valid, the model is still pushed toward an artificially narrow conditional distribution (as we briefly discuss in Sec. 3.1).
>
> This matters beyond retrieval. In generative or instruction-tuned MLLMs, training on a single observed response can suppress other valid responses and reduce calibrated diversity. Recent work, such as DARE [4], also suggests that strong MLLMs (e.g., GPT4, Gemini) still struggle once evaluation explicitly allows multiple correct answers, i.e., when multiplicity is introduced in the benchmark. We will strengthen this discussion in the revision.
>
> - [4] DARE: Diverse Visual Question Answering with Robustness Evaluation
>
> **[Q1] Fundamental principles or guiding strategies**
>
> We agree with the reviewer’s point. It is very difficult to achieve a perfect model or a perfect dataset. In the current submission, we provide actionable strategies to different stakeholders. Our main principle is simple: even if multiplicity cannot be resolved perfectly, it should be taken into account when designing datasets, methods, and benchmarks. Concretely, the guiding principles we advocate are:
>
> - (i) moving from single-positive to multi-positive / ranking-sensitive evaluation,
> - (ii) treating alignments as latent or set-valued rather than strictly one-to-one,
> - (iii) developing representations that capture uncertainty and multiplicity, and
> - (iv) filtering underspecified instances during dataset construction.
>
> **[Q2] Missing citations**
>
> Thanks for the references. These papers are relevant as evidence that downstream VLM adaptation often benefits from representing a class or concept with multiple prompts / distributions, which is consistent with our broader claim that multimodal semantics are often non-unique. We will add this discussion to the revised paper.

---

> > ### Author Rebuttal · Reviewer_sQLR · 2026-04-02
> >
> > I thank the authors for their response. While I agree that the multiplicity issue is indeed a significant challenge in LLMs, and that this position paper aims to highlight it from the perspectives of datasets, training, and benchmarking, the paper unfortunately falls short in providing fundamental principles or concrete guiding strategies, offering only several high-level suggestions. I acknowledge that addressing this issue is inherently difficult at the current stage. Nevertheless, taking all factors into consideration, I am unable to increase my rating and will maintain my original score as Accept.

---

> > > ### Author Response · Authors · 2026-04-04
> > >
> > > We thank you again for the positive feedback and for the very helpful suggestions which we'll incorporate into the next revisions
> > >
> > > If there are any aspects that you think could further improve the clarity, positioning, or impact of the paper for the final version (or any future work), we would greatly appreciate your suggestions.
> > >
> > > Specifically, we would appreciate further clarification on your expectations regarding “fundamental principles”. Our intention was to provide concrete fundamental principles and guiding strategies in the call-to-action section. If they were perceived as high-level to the reviewer, it would be very helpful for us to understand what form of principles or level of concreteness you would find most compelling in this context.

---

### Decision · Program_Chairs · 2026-04-30

**Decision:**

Accept (regular)

**Comment:**

This position paper received four reviews with unanimous Accept ratings (3x Accept, 1x Borderline Accept). All reviewers recognized the value of highlighting multiplicity as a fundamental challenge in multimodal learning affecting the entire pipeline.Reviewers appreciated the articulation of how many-to-many correspondences arise from intra-modal variability, representational asymmetry and task-dependent ambiguity (NoQD, XyCk, sQLR), and the comprehensive coverage across datasets, training, and evaluation (NoQD, FFsh).

Initial concerns included limited empirical grounding (XyCk, sQLR), novelty relative to prior work on modality gaps (FFsh) and applicability beyond retrieval settings (XyCk, NoQD). The authors provided substantial clarifications citing concrete evidence from prior work and explaining how their contribution unifies separate phenomena under a coherent structural property. All reviewers acknowledged their concerns were adequately addressed. The paper is recommended for acceptance as a position paper identifying an important structural challenge deserving the community's consideration in multimodal learning research.